# Student and Supervisor Perspective on Undergraduate Research in a Teaching-Intensive Setting in Oman

**Zainab Al Ajmi, Aaya Al Na'abi, Abdul-Hakeem Alrawahi, Muna Al Saadoon, Huriya Darwish Al Balushi, Fatema Alhabsi and Hamza A. Babiker \***

Department of Child Health, College of Medicine and Health Sciences, Sultan Qaboos University, Muscat 111, Oman
* Correspondence: hbabiker@squ.edu.om

**Abstract:** Undergraduate research (UGR) is a valuable experience that can potentially enhance the quality of graduates, and raise awareness of the importance of research and its impact on career development. These outcomes, however, depend on the environment under which students conduct research. The present study assessed the staff and students' perspective of UGR in an intensive teaching setting at the college of Medicine and Health Sciences (COMHS), Sultan Qaboos University (SQU), Oman. We assessed the perception of supervisors ($n = 90$) and students ($n = 314$) of UGR and factors that hinder the research experience. Satisfaction towards UGR among supervisors and students was good (mean = $72.4 \pm 13.0$) and moderate (mean = $57.8 \pm 14.2$), respectively. The students reported a good satisfaction towards the relevance of UGR (mean = $71.34 \pm 20.0$), the research skills acquired ($63.43 \pm 18.9$), and interaction with research supervisors ($68.47 \pm 23.5$). Female students were more positive towards UGR than males. The students' grade in the UGR module was the only independent factor influencing their satisfaction. Similar to the students, supervisors were highly satisfied with the relevance of UGR (mean = $84.4\% \pm 20.7$), the module structure (mean = $73.3 \pm 14.6$), workload (mean = $73.3 \pm 14.6$) and the students' performance ($71.8\% \pm 18$). However, supervisors were less positive about the students' acquired skills (mean = $69.0\% \pm 12.8$) and available logistics to support UGR (mean = $67.8\% \pm 16.3$). In summary, supervisors and students in COMHS, Oman, (SQU) regard UGR as valuable and recognize its relevance. Supervisors were more likely than students to report a lack of resources to run UGR. Thus, resources should be maintained to inspire supervisors and sustain an active research environment to inspire students.

**Keywords:** supervisors and students' perception; undergraduate research; College of Medicine and Health Sciences; Oman



## 1. Introduction

There is overwhelming evidence supporting the value of undergraduate research (UGR) as an enriching teaching experience [1–3]. UGR helps students to acquire and improve many skills and principles to become active learners and research consumers. Students can, therefore, build up their scientific research knowledge, master basic laboratory skills, and develop research competencies such as critical thinking, problem solving, results interpretation and data communication, both orally and in writing [4]. In The Gulf Cooperation Council (GGC) countries, the value of UGR is recognised by many universities, and different types of UGR programs, including summer research internships and course-based research modules, have been widely integrated into the medical curricula in many medical schools in the region, including in Saudi Arabia [5], the UAE [6], Kuwait, Qatar, Bahrain and Oman [6–8], In addition, there is considerable variation in the duration of the UGR experience, ranging between 4 and 6 weeks in summer internships in universities such as Qatar University (Qatar) and New York Abu Dahbi (UAE), up to >one year at the Sultan Qaboos University (Oman), as it is introduced at different stages (early year vs. late

year [9]). The above differences can results in variation in the perception of students of UGR in different universities in the region [10–13].

Nonetheless, in general, there is a unanimous positive attitude of medical students in the region towards the medical research and its value [14].

In Oman, the College of Medicine and Health Sciences (COMHS), Sultan Qaboos University (SQU), has integrated a course-based undergraduate research module into the medical curriculum since 2009. The major emphasis of the module is on the principles of the research process, including how to define a research problem, conduct literature reviews, collect and analyse data, and write and discuss the results in a research thesis. By the end of 2021, 1655 students had completed the module successfully, and some of the research projects had resulted in publications in international journals. The success of the course-based undergraduate research in COMHS has inspired the medical schools in Oman, e.g., College of Medicine, National University, to adopt this module. The widespread implementation of undergraduate research-based courses in many universities in the country has led The Research Council of Oman to initiate an undergraduate research-funding scheme (https://www.trc.gov.om/trcweb/topics/research/programs/7162, accessed on 28 February 2023). The scheme aims to promote UGR culture in the sultanate. However, no systematic analysis has been carried out to assess how the undergraduate students in Oman perceive UGR or what direct benefits they report.

Many challenges can compromise the quality of the UGR experience and impact the engagement of the students: (i) supervisor-related challenges, such as lack of supervisory skills and research knowledge, and aptitude [15]; (ii) student-related challenges, such as difficulties in managing UGR with other courses, lack of commitment and motivation towards UGR [16,17]; and (iii) institution-related challenges including lack of research capacity, lack of availability of teaching staff to support and engage students in meaningful research driven activities and absence of financial support for research material [15,17]. The above limitations can be more severe in universities with heavy teaching loads and a lack of a strong research culture [17]. A recent study has examined the factors that limit UGR among 228 medical students from different GCC countries: Saudi Arabia (38%), UAE (20.6%), 17.1%, Oman (17.1%), Kuwait (12.7%) and Bahrain (11.4%). The students identified many barriers including time constraints (40 to 61%), as well as lack of mandatory courses on research (39.1 to 57.9%), financial support, statistical support (20 to 78.9%), and mentorship (40 to 63.2%) [7]. Likewise, a survey of 401 medical students in six universities in Saudi Arabia (one private and five public) recognized time constraints (49.9%), lack of formal research methodology training ($n = 170$; 42.4%) and lack of research mentors (23.7%) as the main hindrances to participation in UGR [18]. Many universities in the Middle East are focused on teaching, which deprives staff from scholarly activities to address local issues. For example, it has recently been argued that the high teaching burden among staff in universities in the GCC countries may undermine the potential for more research output, and the development of an adequate research culture [19]. The prevailing environment in such universities may undermine the awareness of the value of creativity and innovation skills' development in undergraduate programs. The key to a successful undergraduate research experience is for students to see and understand the importance and value of responsible research conduct [20].

A preliminary survey carried out In 2016 among the COMHS, SQU, students who completed the UGR module identified some barriers that hinder the research experience [21]. Here, we extended the abovementioned study and investigated both the students' and supervisors' satisfaction and motivation to gain insight into how the UGR experience can be enhanced in a teaching-intensive university. Understanding what students and staff think about UGR, and the factors that limit engagement, will help streamline the delivery of this important module and advise other universities in Oman and the region on effective methods for the delivery of UGR.

## 2. Material and Methods

Study site and subjects. This cross-sectional study was conducted between 2018 and 2019 in COMHS, SQU, the only public governmental medical college in Oman. The UGR module is a course-based research mentored by experienced staff, and it spreads over three semesters of Phase II (3rd year and 4th year) of the MD program. In the first course, in the 2nd semester of the 3rd year, students are introduced to the principles of research method, research methodology and data analysis via a series of lectures and tutorails. At the same time, the students identify a research topic, formulate a research question, and develop outlines of a research proposal. In the second course, in the 1st semester of year 4, the students undertake the research with data collection through surveys and laboratory experiments in line with the research proposal. In the third course, in the 2nd semester of the 4th year, students write up the research data in the form of a thesis and poster.

The present study assessed the perception of 5th and 6th year medical students ($n = 314$) who had completed the three UGR courses between 2017 and 2019. In addition, we evaluated the perception of supervisors ($n = 90$) in 18 different departments in the COMHS and SQUH involved in this module in those three years (2017–2019). The majority of supervisors were from the Departments of Surgery, Medicine, and Microbiology and Immunology, together representing 31.5% of all respondents who completed the study. The lowest proportion of responders was from the Departments of Genetics, and Allied Health Science (Supplementary Table S1). This variation reflects the differences in the number of staff in different departments.

Data collection. Two questionnaires were used in this study, the first for assessment of the students' and the second for supervisors' perception. The questionnaires were developed based on a preliminary study with similar objectives to the present study [21], which developed a study tool based on focused group discussion among a sample of clinical and pre-clinical students who had completed the UGR in COMHS, SQU, between 2015 and 2016. In addition, the preliminary questionnaire was subjected to subject matter expert review [21].

The students' questionnaire comprised two sections: (a) demographics (age, gender, grade (F to A), residence and living conditions; and (b) questions that reflect the perception/satisfaction of students towards UGR. The second section was ordered in six domains: (i) relevance of research to medical students (ii) research supervisor (iii) research skills and learning skills, (iv) supporting resources, (v) time and workload and (vi) teaching and course structure (Supplementary Table S1).

Similarly, the supervisors' questionnaire contained two sections: a) demographic data (age, gender, medical specialty, years of experience, years of involvement in UGR supervision, number of students supervised, attended supervisors' briefing workshop, hours meeting per week and awareness of UGR module). This allows the testing of a possible association between the supervisors' perception and their experience and supervision skills. The second section consisted of six domains: (i) teaching and course structure, (ii) time and workload, (iii) research skills and learning skills, (iv) performance of previous students, (v) supportive resources and materials, and (vi) relevance of undergraduate research.

The questionnaires used in the current study were further subjected to content validation, reviewed by subject matter experts, and piloted on a small group of participants. Items with a good content validation ratio (>0.9) were preserved in both questionnaires. The Cronbach alpha for the student questionnaire ranged from 0.79 to 0.94 (Cronbach alpha for different domains: 0.92 for course structure, 0.79 for time and workload, 0.92 for skills, 0.94 for supervisor, 0.85 for resources, and 0.88 for relevance). For the supervisor questionnaire, the Cronbach alpha ranged from 0.74 to 0.94 (Cronbach alpha for different domains: 0.93 for course structure, 0,74 for time and workload, 0.87 for skills, 0.89 for students, 0.76 for supportive resources, and 0.94 for relevancy).

The two validated questionnaires were self-administered and disseminated through e-mails, WhatsApp messages and hard copies. This helped increase the response rate, which is a known challenge to survey research. The students' and supervisors' e-mails

and phone numbers were obtained from the administration of the COMHS. All data were gathered by the students and staff involved in this research project.

Ethical approval for the study was granted by the Ethics Committee, COMHS, Sultan Qaboos University, Oman (Ref SQU-EC/179/18: MREC 1980).

Data analysis. Data were entered and analysed using the SPSS program Version 23. The qualitative categorized variables were represented as frequency and percentages using frequency tables, pie charts, and bar charts. Continuous variables were presented as mean, standard deviation and median. The total and domain satisfaction score was totalled and divided by the total maximum score and then calculated out of 100%. In addition, we categorized the satisfaction according to cut-off points as poor (<50%), moderate (50–70%) or good (>70%). To assess the univariate associations between satisfaction categorized levels (good/moderate/poor) and factors, Chi-square was used. For multivariate adjusted analysis, we used logistic binary regression. A *P* value of <0.05 was considered significant.

## 3. Results

The perception of UGR was valuated among 314 students and 90 supervisors, in COMHS, SQU, Oman. Then, potential factors that hinder the engagement of students and supervisors with UGR were assessed.

### 3.1. Study Subjects

Out of the 379 targeted students, 314 (82.8%) responded, including 128 (40.8%) male and 186 (59.2%) female, from all governorates in Oman, with exception of Al Wusta (Table 1). The majority (43.9%) of the participants achieved a cumulative grade percentage average (cGPA) of >3, while 8% scored between 2.00 and 2.5 and no students scored < 2.0. However, 72 (23%) students were under academic probation. The vast majority of the students scored A (44, 14.1%) or B (171, 54%) in the final UGR course, while 79 (25.2%) scored C (Table 1).

**Table 1.** Demographic characteristics of medical students in COMHS, Sultan Qaboos University (SQU), involved in the study.

| Characteristics | | Number (%) |
|---|---|---|
| Gender | Male | 128 (40.8%) |
| | Female | 186 (59.2%) |
| cGPA | ≥3.50 | 35 (11.1%) |
| | 3.00–3.49 | 103 (32.8%) |
| | 2.50–2.99 | 151 (48.1%) |
| | 2.00–2.49 | 25 (8.0%) |
| | ≤2.00 | 0 (0%) |
| Being under academic probation | Yes | 72 (23.0%) |
| | No | 242 (77.0%) |
| Future plan to become engaged in research | Yes | 285 (90.7%) |
| | No | 29 (9.3%) |
| Final grade in the research module | A | 44 (14.1%) |
| | B | 171 (54.4%) |
| | C | 79 (25.2%) |
| | D | 19 (6.0%) |
| | F | 1 (0.3%) |

**Table 1.** *Cont.*

| Characteristics | | Number (%) |
|---|---|---|
| Residency regions | Muscat | 84 (26.8%) |
| | South Batinah | 38 (12.1%) |
| | North Batinah | 64 (20.4%) |
| | Dakhliyah | 59 (18.8%) |
| | South Sharqiah | 14 (4.5%) |
| | North Sharqiah | 12 (3.8%) |
| | Musandam | 3 (1.0%) |
| | Dahirah | 25 (8.0%) |
| | Burimi | 11 (3.5%) |
| | Dhofar | 4 (1.3%) |
| | Wusta | 0 (0%) |
| Living condition | Alone | 109 (34.7%) |
| | With roommate | 93 (29.6%) |
| | With parents | 92 (29.3%) |
| | With other relatives | 20 (6.4%) |

We invited 90 supervisors from different departments in COMHS and SQUH involved in UGR between 2017 and 2019. The overall response rate of the supervisors was 96.7% (87/90), including 55 (63.2%) male and 32 (36.8%) female. The mean number of years of total experience of the supervisors was $18.2 \pm 7.7$, and a mean of $4.5 \pm 3.4$ years' experience in UGR supervision. The majority (62%) of the participants were aware of the structure and objectives of the UGR module (Table 2).

**Table 2.** Characteristics of supervisors involved in undergraduate research in COMHS, Sultan Qaboos University (SQU), and Sultan Qaboos University Hospital (SQUH).

| Characteristics | n (%) |
|---|---|
| Gender | |
| Male | 55 (63.2%) |
| Female | 32 (36.8%) |
| Years of experience | |
| 1–10 | 17 (19.3%) |
| 11–20 | 41 (47.0%) |
| 21–30 | 26 (30.1%) |
| >30 | 3 (3.6%) |
| Mean ± SD: | $18.2 \pm 7.7$ |
| Years of involvement in UGR supervision | |
| 1–5 | 58 (66.7%) |
| 6–10 | 26 (29.8%) |
| >10 | 3 (3.6%) |
| Mean ± SD: | $4.5 \pm 3.4$ |
| Number of students supervised | |
| 1–5 | 51 (59.0%) |
| 6–10 | 20 (22.9%) |
| >10 | 16 (18.1%) |
| Mean ± SD: | $8.5 \pm 14.8$ |

**Table 2.** *Cont.*

| Characteristics | n (%) |
|---|---|
| Attended supervisors' briefing workshop | |
| Yes | 39 (45.2%) |
| No | 48 (54.8%) |
| Hours meeting per week | |
| 1–5 h | 55 (63.0%) |
| 6–10 h | 15 (17.3%) |
| >10 h | 17 (19.8%) |
| Mean $\pm$ SD | 7.2$\pm$ 8.1 |
| Awareness of UGR module | |
| Good | 54 (62.1%) |
| Moderate | 27 (31.0%) |
| Poor | 6 (6.9%) |

*3.2. Students' Perception*

The overall students' satisfaction towards UGR was 57.8 (SD $\pm$ 14.2), with a minimum of 20% and a maximum of 92%. The majority of the students were positive about UGR, with 243/314 (77.4%) students claiming a positive (good plus moderate) perception, while 71/314 (22.6%) stated poor satisfaction (Table 3).

**Table 3.** Perception of medical students in COMHS, Sultan Qaboos University (SQU), Oman, of the UGR module.

| Domain | Mean Satisfaction ($\pm$SD) | Min | Max | Good n (%) | Moderate n (%) | Poor n (%) |
|---|---|---|---|---|---|---|
| Relevance of UGR | 71.3 ($\pm$20.0) | 10.3% | 67.0% | 212 (67.5%) | 96 (22.0%) | 33 (10.5%) |
| Research supervisor | 68.5 ($\pm$23.5) | 20.1% | 56.0% | 176 (56.0%) | 75 (23.9%) | 63 (20.1%) |
| Research skills and learning skills | 63.4 ($\pm$18.9) | 21.0% | 42.4% | 133 (42.4%) | 115 (36.6%) | 66 (21.0%) |
| Supporting resources and materials | 55.9 ($\pm$19.0) | 23.2% | 42.7% | 73 (23.2%) | 135 (43.0%) | 106 (33.8%) |
| Time and workload | 54.9 ($\pm$18.3) | 20.0% | 100% | 65 (20.7%) | 137 (43.6%) | 112 (35.7%) |
| Teaching and course Structure | 49.4 ($\pm$15.0) | 8.9% | 50.0% | 28 (8.9%) | 129 (41.1%) | 157 (50%) |
| Overall satisfaction | 57.8 ($\pm$14.2) | 20.0% | 92.0% | 68 (21.7%) | 175 (55.7%) | 71 (22.6%) |

Min = Minimum. Max = Maximum.

We explored the students' satisfaction towards six domains indicative of perception of UGR. Good satisfaction was noted for (i) relevance of research (71.34 $\pm$ 20.0), (ii) research supervisor (68.47 $\pm$ 23.5) and (iii) research skills and learning skills (63.43 $\pm$ 18.9) (Table 3). Moderate satisfaction was reported towards (iv) supporting resources and materials (55.91 $\pm$ 19.0) and (v) time and workload (54.92 $\pm$ 18.3). However, the students were less satisfied with (vi) teaching and course structure (49.36 $\pm$ 15) (Table 3).

*3.3. Factors Influencing Students' Satisfaction*

Table 4 shows univariate analysis of factors that impact students' satisfaction towards UGR. Male students (45.6%) were less positive towards UGR compared to females (54.4%), with $p = 0.048$. With regard to this, compared to females, males seemed significantly less positive in the following modules: time and workload ($p = 0.049$), and research and learning skills ($p < 0.001$).

**Table 4.** Factors influencing the satisfaction of medical students in COMHS, Sultan Qaboos University, Oman, with the UGR module.

| Factors Influencing the Satisfaction | | Good Satisfaction (%) | Univariate *p*-Value |
|---|---|---|---|
| Grade in the last UGR course | A | 75 (23.8%) | <0.001 |
| | B | 219 (69.8%) | |
| | C | 20 (6.3%) | |
| | D | 0 (0.0%) | |
| | F | 0 (0.0%) | |
| Living condition | Alone | 88 (27.9%) | 0.01 |
| | With roommate | 60 (19.1%) | |
| | With parents | 143 (45.6%) | |
| | With other relatives | 23 (7.4%) | |
| Gender | Male | 143 (45.6%) | 0.048 |
| | Female | 171 (54.4%) | |
| cGPA | >3.50 | 32 (10.3%) | 0.14 |
| | 3.00–3.49 | 148 (47.1%) | |
| | 2.50–2.99 | 120 (38.2%) | |
| | 2.00–2.49 | 14 (4.4%) | |
| Being under academic probation | Yes | 65 (20.6%) | 0.19 |
| | No | 249 (79.4%) | |
| Future intension for research engagement | Yes | 291 (92.6%) | 0.71 |
| | No | 23 (7.4%) | |

Grades in the UGR module were positively associated with the satisfaction ($p < 0.00$): students with A (23.8%) and B (69.8%) grades were more positive than those with C (6.3%) and D (0.00%) grade. However, cGPA ($p = 9.14$) and being under academic probation ($p = 0.19$) or intension of considering research in future career ($p = 0.71$) were not associated with the perception of UGR (Table 4).

Unexpectedly, students' living conditions were associated with their perception of UGR. Students living with parents (45.6%) or alone (27.9%) were more inspired than those living with relatives (7.4%) or with a roommate (19.1%) ($p = 0.01$) (Table 4). However, multivariate analysis showed that the grade in UGR is the only independent factor for total satisfaction, with a *p* value < 0.001.

*3.4. Supervisors' Perception towards UGR*

Supervisors showed overall good perception of UGR, with a mean satisfaction of (72.4 ± 13.0), a minimum of 40% and maximum of 98.4%. The majority of participants were either highly (64.4%) or moderately satisfied (29.9%), while only 5.7% were poorly satisfied (Table 5).

Table 5 shows the supervisors' perception of six domains related to UGR. The supervisors were highly satisfied with (i) the relevance of the UGR (84.4% ± 20.7), (ii) the module structure (73.3 ± 14.6), (iii) the workload (73.3 ± 14.6) and (iv) the students' performance (71.8% ± 18). There was a relatively lower perception of (v) the skills that UGR provides the students (69.0% ± 12.8) and (vi) the logistics for UGR (67.8% ± 16.3) (Table 5).

However, univariate analysis showed no association between the characteristics of supervisors, including, gender, total years of experience, years of research supervision, number of students supervised, attending training workshops, time spent on research activities per week and awareness of UGR module, and the total perception of UGR (Table 6).

**Table 5.** Satisfaction of supervisors with UGR in COMHS, Sultan Qaboos University, and Sultan Qaboos University Hospital, Oman.

| | Mean ± SD | Min | Max | Good n (%) | Moderate n (%) | Poor n (%) |
|---|---|---|---|---|---|---|
| Structure | 73.3% ± 14.6 | 36.9% | 100% | 55 (63.2%) | 26 29.9% | 6 (6.9%) |
| Time/workload | 73.3% ± 14.6 | 36.9% | 100% | 55 (63.2%) | 26, 29.9% | 6, 6.9% |
| Skills | 69.0% ± 12.8 | 40.0% | 100% | 44 (50.6%) | 39, (44.8%) | 4, 4.6% |
| Students | 71.8% ± 18.0 | 25.0% | 100% | 61 (69.9%) | 18, 20.7% | 8, 9.3% |
| Supportive resources and materials | 67.8% ± 16.8 | 25.0% | 100% | 47 (54.0%) | 31, 35.6% | 9, 10.3% |
| Relevance of research | 84.4% ± 20.7 | 20.0% | 100% | 73 (83.9%) | 7 (8.0%) | 7, 8.0% |
| Overall satisfaction | 72.4% ± 13.0 | 40.0% | 98.4% | 56, (64.4%) | 26, (29.9%) | 5, 5.7% |

Min = Minimum. Max = Maximum.

**Table 6.** Factors affecting supervisors' perception.

| Supervisors' Characteristics | | Good Satisfaction (%) | Univariate *p*-Value |
|---|---|---|---|
| Gender | Male | 67.3% | 0.4 |
| | Female | 59.4% | |
| Years of experience | 1–10 | 68.8 | 0.1 |
| | 11–20 | 56.4 | |
| | 21–30 | 76 | |
| | >30 | 66.7 | |
| Years of research supervision | 1–5 | 64.3 | 0.6 |
| | 6–10 | 64 | |
| | >10 | 33.3 | |
| Number of students supervised | 0–5 | 63.3 | 0.6 |
| | 6–10 | 57.9 | |
| | >10 | 66.7 | |
| Attending training workshops | Yes | 67.6 | 0.6 |
| | No | 57.8 | |
| Time spent on research activities per week | 0–5 hrs | 70.6 | 0.09 |
| | 6–10 hrs | 50 | |
| | >10 hrs | 50 | |
| Awareness of UGR module | Good | 63 | 0.2 |
| | Moderate | 74.1 | |
| | Poor | 20 | |

## 4. Discussion

The vast majority of undergraduate students (mean 77.4 ±14.2) and supervisors (mean 72.4 ± 13.0), in COMHS, SQU, Oman, expressed a positive opinion of UGR. Female students were more positive than their male counterparts. The final grade of the UGR module was the only independent factor that may influence the overall students' perception. The students were less positive regarding the course structure and supervisors were less satisfied with the students' acquired skills (mean = 69.0% ± 12.8) and logistics for UGR (mean = 67.8% ± 16.3).

The positive perception of SQU medical students towards UGR displays a high awareness of the advantages of research experience and potential future career benefits. This is consistent with the fact that some students tend to put more effort in to complete their data analysis and prepare it for publication. Over the years, undergraduate research in COMHS SQU has resulted in many publications in different medical journals. Such a positive attitude towards UGR is in line with that reported in many medical schools in the region. However, there is a slight variation in the rate of satisfaction in some universities, e.g., King Saud University, Saudi Arabia (97.1%) [10], Ain Shams University, Egypt (74%) [11], the American University of Beirut, Lebanon (4.35 out of 5) [12], and the University of Medical Sciences and Technology, Sudan (91.3%) [13]. The abovementioned inconsistencies in the rate of satisfaction can reflect variability in research facilities and culture. For example, the high perception of UGR at King Saud University, Saudi Arabia, is in line with its high ranking and good research outputs, being ranked as the best university in the region (Scimago institutions ranking, 2022, https://www.scimagoir.com/institution.php, accessed on 28 February 2023). The research capacity in universities in the region varies substantially; the teaching-intensive culture and large intakes can negatively influence the research activities and output [20]. This contradicts the common environment in universities in the West, where research and teaching are well-integrated into the curriculum of research-intensive universities. The variation in the students' perception to UGR has also been linked to the duration of experience and entry time; some studies suggest correlation between the length of time spent in UGR and the benefits [22]. In addition, these differences can result from variability in the year of participation, as the perception of research changes as the student becomes involved in research [4]. One more possible reason for the inconsistent findings is the variability in the scale and cut-off values used in different studies. Nonetheless, in general, there is a unanimous positive attitude of medical students in the region towards medical research and its value [14].

The present study showed an association between the grade achieved in the UGR model and the perception of research: students with a high GPA were more positive than those with lower grades (Table 4). This is in line with a meta-analysis of 37 studies (1,042,537 participants), which showed a relationship between attitude toward research in science and learning achievement [23]. Students with a higher GPA tend to be more confident in their educational experience and achievements [24]. Consequently, they become more inspired for a successful future professional career, and thus possess a good perception of the value of research. However, a study that examined senior high school students showed no significant association between grade level and attitude towards research [25]. Yet, high school students' perception can change once they become engaged with a university program of choice. Some studies have shown that attitude towards science and scientific achievements could vary across different grade levels [24]. In accordance with this trend, it has been suggested that medical students' attitude towards research progressively builds up as the students proceed through the medical program [26]. Thus, a clear career orientation at early stages of the medical school may raise the students' awareness of the value of UGR and its career relevance, thus leading to a more positive attitude and research experience.

However, a large proportion of the students in the present study were less satisfied with the teaching and course structure. Since the initiation of the UGR module in COMHS, SQU, in 2009, there have been regular adjustments to improve the student research experience. This includes organizational guidelines, overall planning, details of timeline for assessment, and assessment rubrics. The content of the taught competences of the UGR module has recently been revised, and some material that overlapped with other courses was removed. Such regular revision of the UGR is critical for engagement and an enriched research experience. The students' perception and engagement are likely to be influenced by the clear organization and logical layout of the course [27].

The good satisfaction of supervisors with UGR (72.4% ± 13.0) can partly be explained by the fact that the staff are familiar with the module, as it has been running for over 10 years

at COMHS. This is reflected in the fact that the supervisors were highly satisfied with the relevance of UGR (mean = 84.4% ± 20.7), the module structure (mean = 73.3 ± 14.6), the workload (mean = 73.3 ± 14.6), and the students' performance (71.8% ± 18) (Table 5). However, supervisors were slightly negative towards facilities and logistics available to run UGR. Thus, supervisors need to be better supported to conduct UGR, with adequate funding and time allocation. The latter has been acknowledged by the COMHS administration, and supervision of UGR has been given a weight of one credit hours per week to faculty involved. UGR supervision is a key factor underpinning the success of the UGR experience; hence, in a teaching-intensive university such as SQU, modality for integration traditional teaching and UGR should be explored [28].

Whilst both supervisors and students regard UGR as a valuable experience, supervisors are more aware of the relevance of UGR (84.4 vs. 71.3) and a lack of resources to run UGR (67.8 vs. 55.9). These findings are expected, as the supervisors are more aware of the value of UGR, and what it takes to run a research project. Interestingly, both supervisors and students are similarly satisfied with interaction with each other (71.8% vs. 68.5%). This contrasts the lack of mentorship reported by students in other universities in the region (7, 19). The good interaction between students and supervisors at COMHS, Oman, can be attributed to clear expectations due to the long-lasting experience of staff with UGR, which has been running for over 10 years.

A limitation of this study is its modest sample size, only including students from COMHS, which may not be representative of students in other universities in Oman and countries in the region. Future research in cooperating multiple medical schools in Oman is needed to assess the UGR perception of medical students. The present study did not examine the different phases of medical education: preclinical, junior and senior clerkship. Some studies identified a more positive attitude toward research courses in medicine connected to more advanced students [26,29]. This study evaluated the UGR module from the students' point of view; however, it is critical to investigate the module efficiency by assessing the students' performance. Therefore, advanced studies, using a large sample size across different medical schools in the region, and utilizing open-ended questions, will provide better insight into the students' views and will allow the development of a sound UGR scheme that will enhance the education experience and the quality of the graduates.

In summary, supervisors and students in COMHS, Oman, (SQU) recognize the value and relevance of UGR. Supervisors were more likely than students to report a lack of resources to run UGR. Supervisors need to be better supported to conduct UGR, and sustain active research environment to inspire students. Further research using mixed methods can explore in more depth the factors influencing the engagement with, or supervision of, UGR.

**Supplementary Materials:** The following supporting information can be downloaded at: https://www.mdpi.com/article/10.3390/educsci13040346/s1, Table S1: Number and percentage of supervisors involved in UGR in different departments in COMHS and SQUH.

**Author Contributions:** Conceptualization, H.A.B. and M.A.S.; methodology, Z.A.A.; validation and formal analysis, A.-H.A., Z.A.A. and A.A.N.; investigation, H.D.A.B. and F.A.; writing—original draft preparation, H.A.B.; writing—review and editing, A.A.N. and M.A.S.; supervision, H.A.B. and M.A.S. All authors have read and agreed to the published version of the manuscript.

**Funding:** This research was supported by the College of Medicine and Heath Sciences, Sultan Qaboos University, Oman. The APC was funded by Deanship of Research, Sultan Qaboos University, Oman.

**Institutional Review Board Statement:** The study was conducted in accordance with the Declaration of Helsinki, and approved by the Ethics Committee, COMHS, Sultan Qaboos University, Oman (Ref SQU-EC/179/18: MREC 1980).

**Informed Consent Statement:** Not applicable.

**Data Availability Statement:** The data presented in this study are available on request from the corresponding author. The data is not publicly available to protect participants confidentiality.

**Conflicts of Interest:** The authors declare no conflict of interest.

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
