# Peer review of "Student and Supervisor Perspective on Undergraduate Research in a Teaching-Intensive Setting in Oman"

_education, doi:10.3390/educsci13040346_

Round 1

Reviewer 1 Report (Previous Reviewer 1)

This looks much better than the previous version, and I have nothing in particular to comment on

Author Response

Response to reviewers

We thank the reviewers for their thoughtful feedback and insight comments. We incorporated most of the suggestions made, in the revised version, those changes are highlighted the manuscript (with track changes). In addition we also attached a clean version (without track changes).

Below we detailed how we have changed the manuscript in line with the reviewers’ suggestions.

Reviewer 1.

  • This looks much better than the previous version, and I have nothing in particular to comment on Suggestions for authors.

Response: We are very pleased that the reviewer finds the revised version of the manuscript well presented, and not needing further comments

Reviewer 2 Report (Previous Reviewer 2)

I have re-reviewed the paper, having read the response from the authors to earlier comments.

Abstract

This was representative of the rest of the paper.

Introduction

This is substantially improved with fewer unsubstantiated statements and gave a good introduction to the purpose of the investigation. It was also more evident what the background was of using research in UG courses. 
The introduction was however fairly long for its actual content.

Methods

The limitations raised earlier persist, as they are central to the work done. This is essentially a questionairre survey, looking at satisfaction in a number of domains, and correlating these with demographic factors. It is clearer now why there was an apparent delay of 1-2 years in collecting this information.
I was confused by the application of Cronbach alpha here. In any instrument one is as interested in correlation between different questions as with divergence between them. The data presented suggests that those who were satisfied in one domain were likely to be satisfied in another domain. However later data presented in the results section spends time dissecting the differences between the domains. It makes no sense to do both. 

Results

These are easy to understand and well presented. Some numbers were bizarre. "Male students (28.1%) were ...". It is not clear what the 28.1% refers to, as it is not the proportion of male students or their positivity.

The take home was well put by the multivariate analysis - those who did well had better satisfaction.

Conclusions

This was quite wordy and did not really do that much to put the findings into context, nor to explore the potential confounders.
The paragraph on female students attempts to put the more positive findings amongst this group into context, but touches on a large and controversial area in gender politics and workplace dynamics. The findings of increased female satisfaction in this study i was hoping would be put in the context of other surveys from the same institution only. This area deserves a full and nuanced discussion, or none at all.

Presentation

The writing remains below an acceptable standard. Despite the authors contention that they have "edited the manuscript throughout and correct all typos and grammatical errors." I'm sure some have been identified, but for an English language journal, there is substantial room for improvement.
Just looking at the first sentence of the abstract, it should be grammatically be re-written from:

"Undergraduate research (UGR) is a valuable experience that can potentially enhance the quality of graduates, and raise the awareness of the importance of research and its impact on career development."

to

"Undergraduate research (UGR) is a valuable experience that can potentially enhance the quality of graduates,  raise the awareness of the importance of research and its impact on career development."

as the authors should not have two 'ands' in a list and should not have a comma before an 'and'. I am not going to point out every grammatical error, that is not my role, nor is it to assess the use of words, but there is a lot of improvement that is still needed here. For instance, two sentences later "The present study assessed the staff and students perspective to UGR in a teaching intense setting..." has two further gramatical errors in addition to poor wording of "teaching intense".

I very much appreciate that the authors are likely to not be writing in their first language. That said, as an English language journal, a good standard of written communication is essential.

There are still some unreferenced acronyms. I might know what GCC and GPA mean, but these still need to be brought out properly in the text.

Overall

The central problem, even after editing, is that the study is essentially a well put together local audit of a local educational initiative, with the only reported outcome being satisfaction.
This is of relevance to this institution, but as reported in the second paragraph of the introduction, this is already well known. 
What would actually be relevant to other institutions and therefore meriting dissemination is useful outcomes: papers produced, impact on careers and the like.

Author Response

Reviewer 2.

I have re-reviewed the paper, having read the response from the authors to earlier comments.

- Abstract

This was representative of the rest of the paper.

Response: We are pleased that the reviewer finds the abstract clearly represents the content of the study

- Introduction

This is substantially improved with fewer unsubstantiated statements and gave a good introduction to the purpose of the investigation. It was also more evident what the background was of using research in UG courses. 
The introduction was however fairly long for its actual content.

Response: We are pleased that the reviewer recognizes the “substantial improvement” of the introduction, and that it provides a  “good introduction to the purpose of the study”. We have removed some references, mistakenly inserted in the text, and this has slightly shortened introduction.

- Methods

The limitations raised earlier persist, as they are central to the work done. This is essentially a questionairre survey, looking at satisfaction in a number of domains, and correlating these with demographic factors. It is clearer now why there was an apparent delay of 1-2 years in collecting this information. 

Response: We are pleased that the reviewer finds the description of the characteristics of the study sample clear.  

- I was confused by the application of Cronbach alpha here. In any instrument one is as interested in correlation between different questions as with divergence between them. The data presented suggests that those who were satisfied in one domain were likely to be satisfied in another domain. However later data presented in the results section spends time dissecting the differences between the domains. It makes no sense to do both. 

Response: The overall Cronbach alpha statistic was deleted from the current version of the manuscript. However, the Cronbach alpha statistic calculated for each domain (reflecting the correlation of the included items in each domain) was reinstated this in the current version (Lines 133-37). 

- Results

These are easy to understand and well presented. Some numbers were bizarre. "Male students (28.1%) were ...". It is not clear what the 28.1% refers to, as it is not the proportion of male students or their positivity.

The take home was well put by the multivariate analysis - those who did well had better satisfaction.

Response: We are pleased that the reviewer finds the results “easy to understand and well presented”.

We have corrected the percentage of male and female students in the text (lines 188-89).

- Conclusions

This was quite wordy and did not really do that much to put the findings into context, nor to explore the potential confounders. 
The paragraph on female students attempts to put the more positive findings amongst this group into context, but touches on a large and controversial area in gender politics and workplace dynamics. The findings of increased female satisfaction in this study i was hoping would be put in the context of other surveys from the same institution only. This area deserves a full and nuanced discussion, or none at all.

Response: We have edited the discussion to put the findings into a clearer context and highlight it implications. We agree with the reviewer that the para on female students “may touch on a large and controversial area” and may not be of direct relevance to the main message of the study. We have deleted this section and related references.

- Presentation

The writing remains below an acceptable standard. Despite the authors contention that they have "edited the manuscript throughout and correct all typos and grammatical errors." I'm sure some have been identified, but for an English language journal, there is substantial room for improvement. 
Just looking at the first sentence of the abstract, it should be grammatically be re-written from:

"Undergraduate research (UGR) is a valuable experience that can potentially enhance the quality of graduates, and raise the awareness of the importance of research and its impact on career development."

to

"Undergraduate research (UGR) is a valuable experience that can potentially enhance the quality of graduates,  raise the awareness of the importance of research and its impact on career development."

as the authors should not have two 'ands' in a list and should not have a comma before an 'and'. I am not going to point out every grammatical error, that is not my role, nor is it to assess the use of words, but there is a lot of improvement that is still needed here. For instance, two sentences later "The present study assessed the staff and students perspective to UGR in a teaching intense setting..." has two further gramatical errors in addition to poor wording of "teaching intense".

I very much appreciate that the authors are likely to not be writing in their first language. That said, as an English language journal, a good standard of written communication is essential.

Response:  We thank the reviewer for pointing to some grammatical errors. We have gone through the whole manuscript and carried out a through editing, with the help of a native English speaker colleague.  

- There are still some unreferenced acronyms. I might know what GCC and GPA mean, but these still need to be brought out properly in the text.

Response: We have checked the whole manuscript and added the full names of all acronyms.

- Overall

The central problem, even after editing, is that the study is essentially a well put together local audit of a local educational initiative, with the only reported outcome being satisfaction. 
This is of relevance to this institution, but as reported in the second paragraph of the introduction, this is already well known. 
What would actually be relevant to other institutions and therefore meriting dissemination is useful outcomes: papers produced, impact on careers and the like.

Response: We thank the reviewer for the above comments/suggestions. The present study provides a valuable insight to UGR in a teaching-intense institute, typical of main universities in the region. We are planning to extended these findings, and evaluate the UGR outcomes, personal and professional gains, and the influence of UGR on career plans. 

Reviewer 3 Report (New Reviewer)

This is an interesting article reporting on perceptions of students and supervisors involved in undergraduate research modules in a medical curriculum at a university in Oman. The article is well structured and presents results in a logical and appropriate way. Some suggestions are provided to enhance the article.

The abstract provides a very good summary of the study. It would help readers to rework the beginning a little to make clear what the gap in understanding is, that you are trying to address.

The version I saw had tracked changes throughout and I was not sure if this was in response to an earlier review?  Or perhaps the submitted manuscript had an error of these changes not being accepted, which made reading a little hard. Please do check this before resubmission. There was also a mixed referencing format – see for example lines 28-33. Please check throughout for consistency of style.

In the introduction I wanted to see more context for the use of UGR in the GGC (and please put in full the first time you use this acronym). You refer to studies but do not elaborate on key findings. This is really important in order to establish the gap you are trying to address with the current study. I wanted to see a more nuanced consideration of what sort of UGR is being employed in these universities and whether there is a mix of types. The term research-oriented is used – just be sure to define this as there are different interpretations. Ron Griffiths (2004, Studies in Higher Education) has a widely used classification, or see work by Mick Healey and Alan Jenkins. See also Beckman and Hensel CURQ 29(4) on ‘Making explicit the implicit: defining undergraduate research’. You comment on line 41 about variations in the perception – what is the nature of these variations?  More context please. I became a little confused about the situation of this study in relation to past research given claim on lines 59-60 that no systematic analyses had been done on the COMHS modules, yet references 6 and 7 suggest maybe there is past research? Perhaps these were from different universities? The fact UGs are getting published in international journals is a fabulous outcome for the programme! The sentence on lines 73-75 needs rewording – I was not sure what was being said.  Remember the literature review should build towards identifying a clear gap in knowledge – this needs some further work so we can see the contribution of this study in relation to past research.

The methods are described quite well, but please add in reference to ethical approval (if there is a number or code). It would really help readers to know about what sort of UGR is being used when (the references suggested above might help).  It was hard to determine the focus (and format in terms of the type of UGR) of the three modules - maybe itemise these. I was unsure how the supplementary tables are being supplied, but trust this will be sorted via a link. When discussing grade (l114) please give the grade range for your GPA (is it 0-4?). On line 120 give all factors rather than using etc.

What a terrific response rate you had for both students and supervisors– well done! The results are presented well, with useful tables displaying appropriate data. It would help readers to introduce the section with a sentence explaining the structure of the results. Can you please check the figures given on L185 as these seem to be different than in Table 4. On l197 5.8% should be 5.7% to match your table. In Table 2 the value for male is missing. In Table 3 there is a mix of decimal places - I suggest standardising to one decimal place for all data in this table (Table 5 is better).

The discussion needs a little more work. I wanted to see a more nuanced consideration of your findings. It was good to see some comparison with past literature, but I wanted to know more about how this complemented and built on earlier research, with a more in-depth exploration of your findings. For example, on line 226 a comment is made about how the “rate of satisfaction can reflect variability in research facilities and culture.”  Can you elaborate to explain in what way and draw on the facilities and culture at your institution (and others) to help explain? A few universities have much higher ratings of satisfaction – why might this be the case? Please elaborate on possible influences in lines 226-231. I was curious to see that female UGs were more positive, but then I noticed that you had a higher percentage of female supervisors – is there a link there perhaps?  Might they have better role models (often we hear about a lack of female role models which can act to the detriment of students). On lines 279-282, please reword as this is a bit awkward. You could say ‘Supervisors need to be better supported to conduct UGR, with better funding and time allocation….’ Can you please explain what 2 credit hours means?  Is this per week or per student per week? I wanted to see some discussion of the differences in ratings for UGR elements between students and supervisors.

Please bolster your summary by restating the aim of the study and the key findings and make clear the original contribution of your research. Then suggest avenues for future study. You might for example suggest a need to use mixed methods and gather some qualitative data to explore in more depth perceptions of engaging in, or supervising, UGR.

The reference list is very good but close attention needs to be given to formatting as many are not correctly formatted.

Minor

Suggest inserting hyphens into course-based and research-based )see l47 and 55 and elsewhere)

Line 56 delete of before ‘an undergraduate…’

Line 72 delete s from undermines

Line 94 insert with after research,

Line 132 insert the after questionnaire and on l137 change increasing to increase

Line 149 insert % after 70

Line 155 and 156 say male and …female and again on l163

L180 reword to females (18.8%; p=0.048).

Line 234 counterparts

Line 240 account for…

Line 267 – unclear meaning – please reword

Line 292-3 repetition of student’s module

Author Response

Reviewer 3

This is an interesting article reporting on perceptions of students and supervisors involved in undergraduate research modules in a medical curriculum at a university in Oman. The article is well structured and presents results in a logical and appropriate way. Some suggestions are provided to enhance the article.

Response: We are very pleased that the reviewer recognises the manuscript is interesting and well presented.

- The abstract provides a very good summary of the study. It would help readers to rework the beginning a little to make clear what the gap in understanding is, that you are trying to address.

Response: We have edited the title to provide a clearer context to the study

- The version I saw had tracked changes throughout and I was not sure if this was in response to an earlier review?  Or perhaps the submitted manuscript had an error of these changes not being accepted, which made reading a little hard. Please do check this before resubmission. There was also a mixed referencing format – see for example lines 28-33. Please check throughout for consistency of style.

Response: The track changes in the previous version were in response to earlier reviewers. Hope the current version, with less track changes, reads clearer. We thank the reviewer for pointing to some references in erroneous format. We have corrected these.

- In the introduction I wanted to see more context for the use of UGR in the GGC (and please put in full the first time you use this acronym).  

Response: We thank the reviewer; we added the full name for all acronyms in the text.

- You refer to studies but do not elaborate on key findings. This is really important in order to establish the gap you are trying to address with the current study. I wanted to see a more nuanced consideration of what sort of UGR is being employed in these universities and whether there is a mix of types.

Response: We thank the reviewer for the suggestion, we have edited the introduction and added more information to highlight the variation in different UGR format in different universities (Lines 34-41), and perceived factors that limit the students’ engagement (lines 69-77).

-The term research-oriented is used – just be sure to define this as there are different interpretations. Ron Griffiths (2004, Studies in Higher Education) has a widely used classification, or see work by Mick Healey and Alan Jenkins. See also Beckman and Hensel CURQ 29(4) on ‘Making explicit the implicit: defining undergraduate research’.

Response: We thank the reviewer for the comments and suggested relevant references. We have now changed the term research-oriented to research active, to clarify the context of the study. 

- You comment on line 41 about variations in the perception – what is the nature of these variations?  More context please. I became a little confused about the situation of this study in relation to past research given claim on lines 59-60 that no systematic analyses had been done on the COMHS modules, yet references 6 and 7 suggest maybe there is past research? Perhaps these were from different universities?

Response: We have added more information to outline the major types and duration of UGR in different universities in region (lines xxxx).  The latter is important as Some studies have documented positive correlations between the length of time spent in URE programs and the benefits to students, e.g.  Jones MT, Barlow AE, Villarejo M (2010). Importance of undergraduate research for minority persistence and achievement in biology. J Higher Educ 81, 82-115

Ref 6 refers to a study carried out in a Kuwaiti private college whereas; Ref 7 examined students from different medical schools, including 29 from Oman, attending a conference. The students from Oman, is a limited and biased sample as they represent student with a positive UGR experience.  

-The fact UGs are getting published in international journals is a fabulous outcome for the programme! The sentence on lines 73-75 needs rewording – I was not sure what was being said.  Remember the literature review should build towards identifying a clear gap in knowledge – this needs some further work so we can see the contribution of this study in relation to past research.

Response: We thank the review for these comments. We have edited the introduction to highlight the gap  and emphasize the rationale.  

- The methods are described quite well, but please add in reference to ethical approval (if there is a number or code). 

Response: We are pleased that the reviewer finds the method section well described. We have completed the information on the ethical approval for the study, and added the approval code (line 147).

-It would really help readers to know about what sort of UGR is being used when (the references suggested above might help).  It was hard to determine the focus (and format in terms of the type of UGR) of the three modules - maybe itemise these.

Response:  We have added extra information to clarify the type of UGR, structure and content of each course of our module (lines 92-100).

-I was unsure how the supplementary tables are being supplied, but trust this will be sorted via a link. When discussing grade (l114) please give the grade range for your GPA (is it 0-4?). On line 120 give all factors rather than using etc.

Response: We than the reviewer for the above suggestions, we have added the grade rage and all supervisors’ characteristics (lines 122-123)

- What a terrific response rate you had for both students and supervisors– well done! The results are presented well, with useful tables displaying appropriate data. It would help readers to introduce the section with a sentence explaining the structure of the results.

Response: We are pleased that the reviewer finds the results section well presented. We added an opening sentence, explaining the structure of the results (lines 158-160).

- Can you please check the figures given on L185 as these seem to be different than in Table 4. On l197 5.8% should be 5.7% to match your table. In Table 2 the value for male is missing. In Table 3 there is a mix of decimal places - I suggest standardising to one decimal place for all data in this table (Table 5 is better).

Response: We thank the reviewer for pointing to the above errors, we have corrected these inconstancies (line 215).

-The discussion needs a little more work. I wanted to see a more nuanced consideration of your findings. It was good to see some comparison with past literature, but I wanted to know more about how this complemented and built on earlier research, with a more in-depth exploration of your findings. For example, on line 226 a comment is made about how the “rate of satisfaction can reflect variability in research facilities and culture.”  Can you elaborate to explain in what way and draw on the facilities and culture at your institution (and others) to help explain? A few universities have much higher ratings of satisfaction – why might this be the case? Please elaborate on possible influences in lines 226-231.

Response:  We thank the reviewer for the interesting comments. We added extra sentences to elaborate on the link between research capacity/culture and engagement of students with UGR (Line 244 to 254).

- I was curious to see that female UGs were more positive, but then I noticed that you had a higher percentage of female supervisors – is there a link there perhaps?  Might they have better role models (often we hear about a lack of female role models which can act to the detriment of students).

Response: Again we than the reviewer for the thoughtful comments. However, we have deleted this section, in response to some suggestion of reviewer 2 that the discussion “touches on a large and controversial area in gender politics”.   

- On lines 279-282, please reword as this is a bit awkward. You could say ‘Supervisors need to be better supported to conduct UGR, with better funding and time allocation….’ Can you please explain what 2 credit hours means?  Is this per week or per student per week?

Response: We thank the reviewer for the above suggestions. We edited the above sentence (lines 298-9), and added information on the weightage of UGR supervision (line 302).

- I wanted to see some discussion of the differences in ratings for UGR elements between students and supervisors.

Response: We than the reviewer for this suggestion, we have added a short para to discuss some differences and similarities between students and supervisors (lines 300-8).

- Please bolster your summary by restating the aim of the study and the key findings and make clear the original contribution of your research. Then suggest avenues for future study. You might for example suggest a need to use mixed methods and gather some qualitative data to explore in more depth perceptions of engaging in, or supervising, UGR.

Response: Again, we have are grateful to the reviewer for good suggestions. We have edited the summary to highlight the major achievements of the study and possible future directions (lines 319-24).

- The reference list is very good but close attention needs to be given to formatting as many are not correctly formatted.

Response: We have edited the references to eliminate inconsistencies in formatting.

- Minor

- Suggest inserting hyphens into course-based and research-based )see l47 and 55 and elsewhere)

Response: Done

Line 56 delete of before ‘an undergraduate…’

Response: Done

Line 72 delete s from undermines

Response: Done

Line 94 insert with after research,

Response: Done

Line 132 insert the after questionnaire and on l137 change increasing to increase

Response: Done

Line 149 insert % after 70

Response: Done

Line 155 and 156 say male and …female and again on l163

Response: Done

L180 reword to females (18.8%; p=0.048).

Response: Done

Line 234 counterparts

Response: Done

Line 240 account for

Response: Done

Line 267 – unclear meaning – please reword

Response: Done

Line 292-3 repetition of student’s module

Response: Done

Reviewer 4 Report (New Reviewer)

Thank you for the opportunity to review the manuscript titled “Undergraduate research in a teaching-intensive setting: student and supervisor perspective” for consideration in Education Sciences.

In this manuscript, the authors employ a cross-sectional study to investigate both students’ and supervisors’ satisfaction and motivation toward undergraduate research. The authors assert that the main goal of this research is to identify barriers that hinder the engagement of students. Although this work is not without merit, as currently presented, it is unclear which of their data provides meaningful insight into their stated research objective.

·      The fourth paragraph in the Introduction provides valuable context on what factors have been shown to compromise the UGR experience. However, the data itself does little to expand on these variables in a way that contributes to what is not known about this phenomenon.

·      The third paragraph of the Introduction provides context on the course but would be better suited under Material and Methods.

·      Overall, the manuscript could be strengthened by refining the Introduction in a manner that more clearly articulate what these data do contribute to the stated objective (Paragraph 1- why is this topic important; Paragraph 2-knowledge gap that you seek to address; Paragraph 3-succinct preview of methods). See Ibrahim and Ghaferi’s chapter on Writing Scientific Manuscripts in HSR. Ibrahim, A.M., Ghaferi, A.A. (2020). Writing Scientific Manuscripts. In: Dimick, J., Lubitz, C. (eds) Health Services Research. Success in Academic Surgery. Springer, Cham. https://doi.org/10.1007/978-3-030-28357-5_22

·      Similarly, the bulk of the Discussion is focused on the overall positive perceptions of both students and supervisors. A more compelling Discussion could be centered around what accounts for the range and variation in the data that were collected. As currently structured, it is unclear what the authors are hoping the reader take away from the data they present.

·      Thorough copyediting of the manuscript is needed.

Author Response

Reviewer 4

- Thank you for the opportunity to review the manuscript titled “Undergraduate research in a teaching-intensive setting: student and supervisor perspective” for consideration in Education Sciences

In this manuscript, the authors employ a cross-sectional study to investigate both students’ and supervisors’ satisfaction and motivation toward undergraduate research. The authors assert that the main goal of this research is to identify barriers that hinder the engagement of students. Although this work is not without merit, as currently presented, it is unclear which of their data provides meaningful insight into their stated research objective.

Response: We are thank the review for the above critique. As detailed in the response to the other reviewers, we have edited different sections in the manuscript,  to clarify that our main aim is to test whether students and supervisors recognise the value of UGR, in an institute where didactic teaching culture is strong.  Having found that both students and supervisors are positive about UGR, we discuss the potential factors that hinder the delivery of UGR.

As mentioned earlier, the findings of the present study, can be extended in a future work to assess the outcomes and impact.  Nonetheless, the data presented here is of interest to other universities in the country/region, that aim to implement UGR.

  • The fourth paragraph in the Introduction provides valuable context on what factors have been shown to compromise the UGR experience. However, the data itself does little to expand on these variables in a way that contributes to what is not known about this phenomenon.

Response: We are pleased that the reviewer finds that the last para in the introduction provide a good rationale t the study. The above piolet data was preliminary, carried out only among students, with a small number of samples and unconfirmed findings.  The present study extended these findings, using a large samples size and tested both students and supervisors, in an institute with strong didactic teaching culture, and potential limitation perceived by both groups.

  • The third paragraph of the Introduction provides context on the course but would be better suited under Material and Methods.

Response: We thank the reviewer or this suggestion, however we feel that this paragraph helps in clarifying the context of the study and the rationale. There are detailed information, in the material and methods section, on the characteristics of our UGR type and the structure.

  • Overall, the manuscript could be strengthened by refining the Introduction in a manner that more clearly articulate what these data do contribute to the stated objective (Paragraph 1- why is this topic important; Paragraph 2-knowledge gap that you seek to address; Paragraph 3-succinct preview of methods). See Ibrahim and Ghaferi’s chapter on Writing Scientific Manuscripts in HSR. Ibrahim, A.M., Ghaferi, A.A. (2020). Writing Scientific Manuscripts. In: Dimick, J., Lubitz, C. (eds) Health Services Research. Success in Academic Surgery. Springer, Cham. https://doi.org/10.1007/978-3-030-28357-5_22

Response: We thank the reviewer for these helpful suggestions, and supporting references, to structure the introduction. We have edited some sections to bring emphasis and highlight the objective and rationale.  

  • Similarly, the bulk of the Discussion is focused on the overall positive perceptions of both students and supervisors. A more compelling Discussion could be centered around what accounts for the range and variation in the data that were collected. As currently structured, it is unclear what the authors are hoping the reader take away from the data they present.

Response: We have heavily edited the discussion to more accurately put our results in the context of the value of UGR and how to improve on the delivery of this important module.

  • Thorough copyediting of the manuscript is needed.

Response: We thank the reviewer for this suggestion, as mentioned earlier we have carefully edited the manuscript guided by the inputs from all reviewers, and we hope the current version is clear enough to follow.

Round 2

Reviewer 2 Report (Previous Reviewer 2)

n/a

Reviewer 4 Report (New Reviewer)

The authors have adequately addressed previous comments. Close copyediting for grammar, clarity, and formatting is required.

This manuscript is a resubmission of an earlier submission. The following is a list of the peer review reports and author responses from that submission.

Round 1

Reviewer 1 Report

Dear author(s)

Thank you for the opportunity to read this manuscript. It is an interesting topic and clearly extensively analysed, but there is a structural problem with contextualising the study. Even though it is clear from the Discussion section that other research has been carried out on such a topic and in the same region, this comes far too late. You need to contextualise the work before the Method section in a Background research section. This will help the reader to understand why the study is important. Also, the conclusions are very sparse... what are the implications of your study for instructors who want to set up an UGR course? These are very important issues that need to be addressed before this can be published.

Good luck with your revisions

Reviewer 2 Report

Overall

The premise of the paper is good - that a new innovation, having been introduced to a course some time ago, is now tested for its impact. It would be fabulous if all such innovations were subjected to proper evaluation.

The relevance

Many courses have research elements, and getting beyond the ideas that it is good because research skills are good, and such courses are popular (perhaps due to research-related Kudos in the medical world) is hard. Transferrable lessons would be valuable to other institutions.

Introduction

The opening statement is disappointing. The "overwhelming evidence supporting the value of undergraduate research" is linked to just one citation, and a citation that looks at characteristics of research blocks rather than the value to the student. The rest of the introduction was better, but it was not easy to see what the intended investigation was about, and why it was needed to be done, and how this would add to the preliminary findings from the same setting "Medical students’ perception towards undergraduate research".

Methods and Approach

Essentially this is a questionnaire survey, with steps to ensure the questionnaire is as good as possible. It was unclear why the survey was not administered until 2-3 years after the course, and also why anything above Kirkpatrick level 2 outcomes were not sought.

Data collection and analysis steps seem appropriate at first reading.

Results

There are a number of substantial problems with the results section. Firstly, the 'satisfaction' is completely arbitrary and merely tells the reader that the student did or didn't like the course, as opposed to finding it useful for their career, enabling a publication, altering patient care etc. Second, finding associations between demographics and such an outcome is also meaningless. Were female students generally more positive in their feedback in other areas of the course for instance? Next there is loose use of language. High levels of satisfaction do not mean the student is inspired, and also an association is not the same as an impact.

There is inconsistent use of acronyms throughout and all should be preceeded by the words in full.

Discussion

This is much longer and better written, however is far too long for the results found. The limitations section would need to include many other issues - retrospective, survey, unclear what % responded, single site and more.

Presentation

The writing was of fair standard, but would need to be improved. There were numerous minor gramatically errors or instances of poor choice of words. Many words were unclear in their meaning in the context. 

Overall

I applaud the authors for attempting to properly evaluate an element of their course. However, this is a very limited analysis of a local course. It would tell the reader that students like teaching, and value learning research skills. This does not help the reader plan a research module in their institution, nor does it tell the reader anything about the value of putting research into a course in terms of the doctors the course produces.